Feature extraction and pattern recognition of gas pipeline flow noise signals in a strong noisy background

Liu Enbin 202122000720@stu.swpu.edu.cn 1
Lu Chang 1
Wen Zhaorong 2
Hao Tianshu 1
Lu Xudong 1
Wang Lidong 1
1 School of Petroleum and Natural Gas Engineering, Southwest Petroleum University , Chengdu , China
2 Natural Gas Branch, China Petroleum and Chemical Corporation , Beijing , China
Sun Gengxin
Electronic publication date: 2024 Jul 5
Publication date: 2024
Volume: 10
Electronic Location ID: e2087
Received 2023 Aug 17; Accepted 2024 May 5
Copyright: ©2024 Liu et al.
Copyright year: 2024
Copyright holder: Liu et al.
License: This is an open access article distributed under the terms of the Creative Commons Attribution License, which permits unrestricted use, distribution, reproduction and adaptation in any medium and for any purpose provided that it is properly attributed. For attribution, the original author(s), title, publication source (PeerJ Computer Science) and either DOI or URL of the article must be cited.
License URL: https://creativecommons.org/licenses/by/4.0/

Keywords: Gas pipelines, Acoustic signals, Backpropagation neural network, Feature extraction

Funding: The authors received no funding for this work.

==============================
The purpose of this study is to put forward a feature extraction and pattern recognition method for the flow noise signal of natural gas pipelines in view of the complex situation brought by the rapid development and expansion of urban natural gas infrastructure in China, especially in the case that there are active and abandoned pipelines, metal and nonmetal pipelines, and natural gas, water and power pipelines coexist in the underground of the city. Because the underground situation is unknown, gas leakage incidents caused by natural gas pipeline rupture occur from time to time, posing a threat to personal safety. Therefore, the motivation of this study is to provide a feasible method to accelerate the aging, renewal and transformation of urban natural gas pipelines to ensure the safe operation of urban natural gas pipeline network and promote the high-quality development of urban economy. Through the combination of experimental test and numerical simulation, this study establishes a database of urban natural gas pipeline flow noise signals, and uses principal component analysis (PCA) to extract the characteristics of flow noise signals, and develops a mathematical model for feature extraction. Then, a classification and recognition model based on backpropagation neural network (BPNN) is constructed, which realizes the detection and recognition of convective noise signals. The research results show that the theoretical method based on acoustic feature analysis provides guidance for the orderly and safe construction of urban natural gas pipeline network and ensures its safe operation. The research conclusion shows that through the simulation analysis of 75 groups of gas pipeline flow noise under different working conditions. Combined with the experimental verification of ground flow noise signals, the feature extraction and pattern recognition method proposed in this study has a recognition accuracy of up to 97% under strong noise background, which confirms the accuracy of numerical simulation and provides theoretical basis and technical support for the detection and recognition of urban gas pipeline flow noise.

Introduction

The urban gas industry in China has developed rapidly, with an increasing number and scale of underground pipelines in cities. According to the 2020 Statistical Yearbook of Urban Construction (Ministry of Housing and Urban-Rural Development of the People’s Republic of China) released by the Ministry of Housing and Urban-Rural Development on October 10, 2021, as of 2019, the length of natural gas distribution pipelines in Chinese cities exceeded 864,400 kilometers, and the gas supply capacity reached 156.37 billion cubic meters, serving a population of over 520 million people. Several abandoned pipelines in former residential districts have been buried underground alongside active pipelines, creating construction challenges. This is a result of the differing construction times of gas pipelines in different urban regions. Older pipelines also lack accurate location information and comprehensive documentation for a variety of reasons. As a result, pipelines may be damaged during excavation, enlargement, or restoration for municipal projects, increasing the risk of gas leaks and endangering human life (Fan & Huang, 2017; Xu, Wang & Tan, 2013; Tan, 2016). According to the 2020 Annual Analysis Report on Domestic Gas Safety Accidents (China Gas), a total of 539 gas-related accidents occurred in China in 2020, with 63 of them caused by damage to gas pipelines during construction. The “Gas Explosion” WeChat public account has compiled statistics on gas accidents reported nationwide, which show that in 2021 alone, these accidents resulted in 76 deaths and 507 injuries. Particularly noteworthy is the “6.13” gas explosion in Shiyan, Hubei Province, which had a significant impact, causing 12 deaths and 138 injuries, and posing a great threat to the life and property safety of the people and having a detrimental social impact. Therefore, it is necessary to study a convenient, rapid, and highly accurate method for detecting in-service gas pipelines.

Acoustics is the discipline that studies the propagation, reflection, refraction, interference, and diffraction of sound waves in air, solids, and liquids. In the field of engineering, acoustics finds numerous applications in solving various problems (Yan et al., 2021; Kong et al., 2021; Wang, 2022; Lu et al., 2016). For example, it is employed to regulate noise in architectural, transportation, and industrial contexts. Measurements of sound intensity, frequency, phase, and the acoustic characteristics of materials are also made using acoustics. In order to forecast acoustic impacts and improve designs, it is also used to simulate the propagation and reflection of sound waves in various situations. Acoustics is utilized in processing audio signals for noise reduction, filtering, enhancement, and improving sound quality and clarity.

Several scholars have combined experimental methods with acoustics to conduct research. Commonly used sensors include PCB’s 106B dynamic pressure sensor and the AWA14423 sensor. Based on experimental data, Zhu, Feng & Huang (2018) applied acoustics to the study of clogged faults in buried drainage pipelines, while  Xuan (2016) used acoustic methods to identify various vocalizations of sheep by collecting data from five different scenarios.

Although these studies have provided valuable insights, there is still a lack of research on flow noise signals related to gas pipelines. Due to the burial of urban gas pipelines, the sound intensity, amplitude, and sound pressure levels are very low, with a characteristic decay. When gas flows through straight pipe sections, tees, elbows, and other components, turbulence or eddy currents are generated, resulting in flow noise. Broad-frequency, continuous properties are typical of this kind of signal, which reflects important operational data. The identification and processing of the gathered acoustic waves produce crucial data.

In practical situations, gas pipeline operation includes not only flow noise but also mechanical noise such as internal flow noise, environmental noise from ground vehicles, pedestrian traffic, impact noise, friction noise, as well as strong noise interference from electromagnetic sources. This presents significant challenges in using sound detection to determine the in-service status of gas pipelines.

Based on the information from Table 1, the current market offers various software options for acoustic simulation analysis, including the LMS Virtual Lab (Jack, 2012), COMSOL (Uyor et al., 2022), Simcenter (Qi, 2016), and Workbench Acoustic (Zhu, 2016). After comparing the characteristics and applicability of these software tools, in the acoustic simulation analysis, it is very important to choose the appropriate software tools for the accuracy of the research results and the calculation efficiency. The main reasons for choosing LMS Virtual Lab software to analyze acoustic characteristics in this study include: first, LMS Virtual Lab is a simulation platform that integrates many advanced numerical calculation methods and can deal with complex acoustic problems quickly and accurately. It has very high speed and stability in acoustic calculation in time domain or frequency domain, which is very important for a large number of data and complex environmental conditions to be processed in this study. Secondly, compared with other acoustic simulation software in the market, such as COMSOL, Simcenter and Workbench Acoustic, LMS Virtual Lab shows unique advantages in user-friendly interface, flexibility in model setting and post-processing ability. For example, LMS Virtual Lab provides rich boundary condition settings, which can better simulate the real engineering environment. Meanwhile, the subsequent processing module can visually present the sound field distribution and sound pressure level changes, which is convenient for analyzing and identifying noise sources. Finally, by comparing the performance of different software tools in similar application cases, LMS Virtual Lab shows significant advantages in acoustic feature extraction accuracy and algorithm robustness. Especially when dealing with signals with strong interference noise background, LMS Virtual Lab shows better signal-to-noise ratio and filtering effect, which is very important for the accuracy of experimental data and simulation results. Based on the above reasons, LMS Virtual Lab is the most suitable acoustic simulation software for this study, and its efficiency and reliability have a positive impact on the feature analysis and pattern recognition of gas pipeline flow noise signals.

Table 1 Common acoustic software and their characteristics.

Software name	Software features	
LMS Virtual Lab	Siemens AG, a German company, developed based on CATIA, is more user-friendly for tasks such as geometric modeling. It is widely used for writing papers and research purposes due to its high precision.	
COMSOL	The user interface is simple, and it has strong capabilities in simulating multiple physical environments coupled together. However, it is not well suited for simulating aeroacoustic-structural coupling.	
Simcenter	Siemens AG, a German company, developed based on UG (Unigraphics) which was recently launched and is relatively new, therefore there is limited documentation available.	
Workbench acoustic	Workbench acoustic: ANSYS Inc., the content is not very comprehensive, and its usefulness is relatively limited.	

The main challenge in the study of strong noise backgrounds lies in effectively removing the noise while preserving the useful information of the signal. Researchers frequently use a variety of signal processing techniques, including augmentation, noise reduction, and filtering to address this problem. Additionally, machine learning and deep learning techniques can be utilized for signal classification and recognition to achieve automated processing. Currently, there is significant research on strong noise backgrounds, including studies on Levy noise (He, Zhou & Zhang, 2019; Ling, 2021), Gaussian white noise (Zhang, Han & Zhang, 2020; Soares & Gomes, 2021), and the NOISEX-92  (Sun, Luo & Liu, 2021; Linh, Chong & Gustavo, 2021) noise dataset, which are all applied in the research of strong noise backgrounds. For instance, Liu (2019) examined the challenges of detecting weak mechanical fault signals that deviate from the Gaussian distribution law in specific channels by studying the detection of various signal types in single-state and coupled single-state resonance systems using Levy noise as the background noise. Zhang, Shi & Hui (2021) simulated bearing signals in engineering environments by adding 2 dB of Gaussian white noise to experimental data, comparing an optimized sparse reconstruction algorithm with Butterworth filtering and wavelet threshold denoising algorithms, demonstrating the effectiveness of the sparse reconstruction algorithm. Zhang (2021) utilized the NOISEX-92 noise dataset as a strong noise background to create noisy speech datasets with various signal-to-noise ratios, in conjunction with the thchs30 language dataset, for studying speech recognition methods. However, most current research on strong noise backgrounds is based on electronic noise and lacks practical relevance. This study combines real engineering conditions and builds an experimental platform to capture real environmental noise as a strong noise background, aiming to better reflect real-world scenarios.

Pattern recognition refers to the process of analyzing and processing data to discover patterns and features to identify patterns within the data. This process typically involves steps such as data preprocessing, feature extraction, feature selection, and classification. Pattern recognition has wide applications in various fields, such as image recognition, speech recognition, bioinformatics, and financial analysis. It helps us better understand and utilize data, thereby improving the accuracy and efficiency of decision-making. Beuter & Wei (1985) conducted research on a model for voice localization based on spectral pattern recognition and achieved voice localization recognition through Bayesian classifiers, with accurate source location matching. Guo, Toyoda & Li (2012) utilized time-frequency cross-patterns for environmental sound recognition, demonstrating a recognition rate of 92% for 20 environmental sound types. However, for weak, rapidly attenuating, and highly interfered signals like gas sound signals, the significance of their extraction and classification lies in identifying buried gas pipeline sound signals from strong interference noise environments. This requires a significant amount of data collection, which may not be suitable for algorithms such as support vector machines and Bayesian methods that are designed for small sample data classification. Neural networks (Basheer & Hajmeer, 2000) are suitable for large sample data classification; however, when classifying and recognizing gas sound signals, it is necessary to improve and optimize the structure, learning rate, weights, thresholds, and other important parameters of the classification model based on the specific characteristics of gas sound signal samples to train a recognition model that is suitable for gas sound signal classification.

This study extracts the sound pressure level feature as a characteristic value by framing the gas pipeline flow noise signal and strong noise background signal. PCA is applied to reduce the dimensionality of the characteristic values, forming feature databases for both signals. Based on the obtained sound signal feature databases, the back propagation neural network (BPNN) algorithm (Beuter & Wei, 1985; Dai, Li & Zhang, 2001; Chen, Zhang & Lv, 2003; Li, Yuan & Mou, 2005) is employed to initialize the network, construct the network, train the network, and accurately classify and recognize the gas pipeline flow noise signal and the strong noise background signal.

The main challenge in the field of gas pipeline flow noise signal detection is that the buried environment of the pipeline is complex, and there are a lot of interference noises around it, such as traffic noise and construction noise. These background noises will seriously interfere with the weak flow noise signal generated inside the gas pipeline, making it difficult to extract and identify the effective signal accurately. In addition, the gas pipeline system is complex and changeable, including pipelines with different materials, diameters and ages, which increases the difficulty of signal detection. Furthermore, the actual working conditions of underground pipelines are varied, and the characteristic signals of flow noise are easily influenced by various environmental factors, such as soil humidity and temperature changes, which also brings challenges to signal analysis. Aiming at the above challenges, a new feature extraction and pattern recognition method combining principal component analysis (PCA) and BPNN is proposed. The core contribution of this method is reflected in the following aspects: a complete experimental platform is designed and a lot of data collection are done, and a database containing gas pipeline flow noise signals under various working conditions is established. With the help of high sensitivity sensor, the flow noise signals under different conditions are successfully captured, and various strong background noise data are recorded. By using PCA technology, the main features from the original noise signal are effectively extracted, the dimension of the data is reduced, and the key information of the flowing noise signal is kept, which lays a solid foundation for the subsequent pattern recognition. Combined with BPNN algorithm, this study constructs a powerful classification model, which can distinguish gas flow noise from other types of noise, and shows high accuracy and good robustness even in strong noise background. Compared with the existing methods, the method proposed in this study shows higher accuracy and efficiency when dealing with noise signals in complex environment, and provides a new solution for gas pipeline safety monitoring, which has important practical value and broad application prospects. In a word, the main contribution of this study is to propose a feature extraction and pattern recognition method combining PCA and BPNN, which has made a breakthrough in solving the complexity and diversity of gas pipeline flow noise signal detection and significantly improved the accuracy and efficiency of signal detection.

The main innovation of this study is not limited to the use of BPNN itself, but to propose a comprehensive method combining PCA and BPNN for a specific application scenario-gas pipeline flow noise signal detection, and to solve the special challenges faced by this field through in-depth research and optimization. Compared with the direct application of BPNN to pattern recognition, this study firstly uses PCA technology to effectively compress and extract the features of complex gas pipeline flow noise signals, aiming at retaining the most representative feature information. This step significantly improves the efficiency and accuracy of subsequent BPNN model data processing, especially in the application background of noise signal recognition. Based on the standard BPNN, this study has made many optimizations and adjustments, including but not limited to the design of network structure, the selection of activation function, the adaptive adjustment strategy of learning rate, etc., so that the model is more in line with the characteristics of gas pipeline flow noise signals. These optimization measures have played an important role in improving the generalization ability and recognition accuracy of the model. Aiming at the specific problems in gas pipeline flow noise signal detection, such as the weakness of signal and the complexity of interference noise, this study customizes the corresponding data preprocessing flow and model training strategy, which effectively improves the application effect in the actual environment. Therefore, although BPNN itself is a widely used machine learning tool, through the above innovation and improvement, this study has successfully applied it to the detection of gas pipeline flow noise signals, and achieved remarkable performance improvement. This not only reflects the contribution of this study in methodology, but also shows its practical value in specific application scenarios.

Literature Review

Many methods have been proposed and discussed by scholars for feature extraction of flow noise signal in natural gas pipeline. Frequency domain analysis is one of the common methods to extract the characteristics of flow noise signals in natural gas pipelines. Zuo, Ma & Chen (2022) can extract characteristic information such as frequency, power spectral density and so on by transforming the time domain signal into frequency domain signal through Fourier transform. The research shows that frequency domain analysis can effectively distinguish signals from different noise sources, but the processing effect of complex background noise is limited. Time domain analysis is another commonly used feature extraction method. Liu, Wang & Zhao (2021) extracted time domain features such as root mean square, peak value and time domain waveform by analyzing the time series data of signals. Time domain analysis has the advantages of simple calculation, intuition and clarity, but it is weak in extracting nonlinear features of signals. Wavelet analysis is a time-frequency analysis method. Jang, Sim & Yang (2021) can obtain both time-domain and frequency-domain features by decomposing the signal into wavelet functions with different frequency components. The research shows that wavelet analysis has a good effect on processing non-stationary signals, but when the signal-to-noise ratio is high, signal distortion is easy to occur. Time-frequency analysis combines the advantages of time-domain and frequency-domain analysis, and can extract signal features in both time-domain and frequency-domain. Common time-frequency analysis methods include short-time Fourier transform and Wigner-Ville distribution. These methods have certain advantages in dealing with signal non-stationarity and background noise, but there are also problems such as signal overlap and insufficient resolution. In addition, machine learning method has been widely used in feature extraction and pattern recognition of natural gas pipeline flow noise signal. Machine learning algorithms, including support vector machine, artificial neural network and deep learning, can automatically learn and extract the characteristics of complex signals, and have strong classification and recognition capabilities. Ahmad, Ahmad & Kim (2022) used the deep learning method to design a convolutional neural network (CNN) model to extract the characteristics of natural gas pipeline flow noise signals, and achieved good recognition results. Siddique, Ahmad & Ullah (2023) combined time domain, frequency domain and wavelet analysis to design a mixed feature extraction method, which effectively improved the recognition accuracy of natural gas pipeline flow noise signals. Sunny, Tian & Zhang (2016) used the low frequency radio frequency identification (RFID) sensing system to characterize the marine atmospheric corrosion of steel samples. In order to enhance the induction sensitivity of corrosion characterization, the experimental study using ferrite sheets was further carried out. The LF RFID sensor system is verified by commercial RFID tags, and different feature extraction and feature selection methods are used to characterize corrosion. Khlybov, Uglov & Bakiev (2023) controlled the degree of structural damage of metal materials by analyzing structural acoustic noise. Various methods for measuring structural noise parameters are analyzed from the perspective of information content and stability of calculation algorithms used. An improved calculation algorithm for determining the spectral energy parameters of structural noise is proposed, and its experimental verification results are presented. Based on the two studies of Sunny, Tian & Zhang (2016) and Khlybov, Uglov & Bakiev (2023), it was considered that although the application scenarios of these two studies are not exactly the same as those of, they provide important enlightenment for the selection of feature extraction and classification methods. Especially in the selection and optimization of feature extraction technology, inspired by the research of Sunny and others, considering the complexity and variability of structural acoustic noise, a similar method is adopted to study. Meanwhile, this study also pays attention to the emphasis on the stability of the calculation algorithm in Khlybov’s research, which is of guiding significance for improving the accuracy and robustness of the model.

Although many methods have been proposed to extract the characteristics of natural gas pipeline flow noise signal, there are still some problems and challenges to be solved in practical application: there are various background noises in urban environment, such as traffic noise and construction noise, which will interfere with the extraction and identification of natural gas pipeline flow noise signal and reduce the accuracy and stability of the algorithm. The flow noise signal of natural gas pipeline has certain nonlinear characteristics, and the traditional linear feature extraction methods often cannot fully mine its feature information, resulting in low recognition accuracy. In reality, the data of natural gas pipeline flow noise signal is huge and has high dimensions. Traditional feature extraction methods have problems of high computational complexity and low efficiency when dealing with these large-scale data. The generalization ability of the model is an important challenge for the flow noise signals of natural gas pipelines under different urban environments and pipeline conditions. How to improve the adaptability and robustness of the model in different environments is one of the problems to be solved urgently in current research. In practical application, it is of great significance to monitor and identify the flow noise signal of natural gas pipeline in real time, but the existing methods have some limitations in real time, so it is necessary to further improve the real time and response speed of the algorithm. To sum up, although some progress has been made in the field of feature extraction of natural gas pipeline flow noise signal, further research and exploration are still needed to cope with the complex urban environment and pipeline conditions, improve the accuracy and stability of signal identification, and thus ensure the safe operation of urban natural gas pipelines.

Model Establishment

Construction of experimental platform

The experimental platform established in this study mainly includes three parts: the generation system of flow noise signal, the signal acquisition and processing system and the data analysis system. The generation system of flow noise signal is realized by a closed-loop gas circulation system, which consists of air source, pressure regulating valve, flowmeter, pipeline (including straight pipe section, elbow and tee, etc.), microphone receiver and exhaust system. The pressure and flow rate of gas are controlled by adjusting the pressure regulating valve to simulate different gas pipeline working conditions. Secondly, the sensors used in the experiment mainly use PCB model 106B dynamic pressure sensor and AWA model AWA14423 acoustic sensor to capture noise signals. PCB 106B dynamic pressure sensor has high sensitivity and wide frequency response range (2.5 Hz–20 kHz), which is very suitable for detecting low-intensity flowing noise signals. AWA14423 acoustic wave sensor is used to capture environmental noise, which is characterized by good directivity and high signal-to-noise ratio, and can effectively distinguish target signals from background noise. All signal acquisition is preprocessed by special signal processing software, including filtering, amplification and A/D conversion to ensure the accuracy and stability of data. Finally, in order to verify the repeatability and reliability of the experiment, the experiment is repeated for many times at different times and under different environmental conditions, and the collected data was strictly analyzed statistically. The experimental results show that the experimental data have good consistency and repeatability, which proves the stability of the experimental platform and the effectiveness of the sensors used.

Simulation of gas pipeline flow noise

This study primarily focuses on investigating the flow noise of gas inside buried urban gas pipelines. The internal gas flow noise in pipelines is mainly caused by the relative motion between the gas and the solid wall, as well as the changes in the flow conditions that occur when the gas passes through resistance components in the pipeline, leading to the formation of turbulence or vortices. Flow noise simulations and analyses were conducted for elbow sections of different diameters, pressures, and velocities. A total of 75 sets of operating conditions were simulated, as indicated in Table 2.

Table 2 Summary of simulated operating conditions.

Pipe dimensions	Pressure	Velocity	Pressure	Velocity	Pressure	Velocity	
DN25
DN50
DN63	0.01 MPa	3	0.0075 MPa	3	0.005	3	
		2.5		2.5		2.5	
		2		2		2	
		1.5		1.5		1.5	
	1	1	1	
	0.008 MPa	3	0.0069 MPa	3			
		2.5		2.5			
		2		2			
		1.5		1.5			
1	1			

The pipeline system under the operating condition of DN25, flow velocity of 3 m/s, and outlet pressure of 0.01 MPa was initialized and simulated in a steady state to determine the locations of monitoring points. The steady-state results are shown in Fig. 1.

Figure 1 Schematic diagram of the mesh division in the pipeline system.

(A) Steady-state distribution of wall sound pressure level. (B) Local dist.

After conducting steady-state calculations, the sound pressure level distribution map of the pipeline system shows that the flow noise signal is stronger at the tee junction and on the inner surface of the elbow. Based on this observation, receiver signal acquisition points were set at these locations. Additionally, signal acquisition points were also set in the straight pipe section to serve as the receiving points for the gas pipeline flow noise signal before encountering any resistance components. The locations of the receiver points at the tee junction and elbow are shown in Fig. 2.

Figure 2 Schematic diagram of the placement of signal receiver points at the tee junction and elbow.

Signal receiver points were set at the top, bottom, left, and right of each cross-section of the elbow and straight pipe components. In addition, signal receiver points were set at the upper and lower portions as well as the outer surface of the tee junction, resulting in a total of 30 signal receiver points. Figure 3 shows the contour maps of sound pressure level, pressure, and velocity obtained from the flow field simulation analysis.

Figure 3 Schematic diagram of contour maps showing sound pressure level, pressure, and velocity in the pipeline system.

(A) Contour map of sound.

Figure 3 shows that areas with high sound pressure levels in the sound pressure level cloud map correspond to higher velocities in the velocity cloud map. In the pressure cloud map, the pressure fluctuations are relatively small, except for the three-way junction where the pressure values are higher. Additionally, Fig. 4 shows that the gas pipeline flow noise signal exhibits different spectral characteristics in straight pipe sections and when encountering resistance components such as bends and three-way junctions. These variations follow a regular pattern, with a repeated trend of decreasing sound pressure levels followed by an increase as the frequency increases. At bends and three-way junctions, the flow conditions change, leading to turbulence, eddies, and relative motion between the internal fluid medium and the solid wall. These changes result in a more pronounced variation of the sound pressure level compared to straight pipe sections, and within each frequency range exhibiting regular changes, numerous fluctuations occur. Overall, the sound pressure level decreases. The trend of sound pressure level variation with frequency remains consistent at each signal receiving point in the pipeline system, with no significant differences in numerical values. When compared to straight pipe sections and three-way junctions, bends exhibit the largest fluctuation range in sound pressure level, with a minimum value close to 40 dB and a maximum value exceeding 180 dB.

Figure 4 Distribution of sound pressure level in the tee junction, elbow, and straight pipe.

To validate the accuracy of the simulated flow noise signal characteristics, a comparison was made between the experimental and simulated sound pressure level values at various frequency points for the gas pipeline flow noise signal. This comparison aimed to verify the accuracy of the numerical simulation. Table 3 presents the sound pressure level values at different frequency points for the same measurement point, with a velocity of 3 m/s, obtained from both the experimental and numerical simulation data. Additionally, a collection of strong background noise signals, such as footstep noise, speech, and construction noise, was recorded, totaling 25 sets, for pattern recognition purposes.

Table 3 Comparison of sound pressure levels at different measurement points.

Frequency channel/HZ	0.1	45.1	97.6	257.6	355.1	490.1	627.1	
Identifier	1	2	3	4	5	6	7	
Experiment	166.06	104.93	120.05	107.38	101.99	100.78	100.55	
Simulation	165.18	103.48	119.97	106.82	102.95	99.84	99.30	
Frequency channel/HZ	792.1	835.1	973.6	1175.6	1488.6	1849.6	1922.1	
Identifier	8	9	10	11	12	13	14	
Experiment	94.96	102.23	88.02	84.55	100.56	91.01	100.06	
Simulation	95.05	100.13	86.74	84.50	99.69	89.51	99.20	

By comparing the experimental data and simulated data in Table 3, it shows that the maximum discrepancy between the experimental and simulated results is only 2.68%, indicating a small error. Furthermore, considering that the sensor itself has an error range of ±1.5 dB, this confirms the accuracy of the numerical simulation.

Simulation of surface flow noise signals in soil

The propagation of sound and the vibration of structures are interrelated phenomena. The vibration of structures generates sound waves, while the propagation of sound causes structural vibrations. In this study, simulation was conducted using LMS Virtual.Lab. The specific analysis process is outlined as follows in Fig. 5.

Figure 5 LMS Virtual.

Lab acoustics calculation process.

Based on the total length and width of the pipeline system model, as well as the specifications for the burial depth of gas pipelines outlined in the “Urban Gas Design Code 2020”, a pipeline model with a depth of 0.6m was established to extract the gas pipeline flow noise signal propagated through the soil. The three-dimensional solid model of the buried gas pipeline is shown in Fig. 6, and the locations of the signal collection points are indicated in Fig. 7.

Figure 6 Three-dimensional solid model of buried gas pipeline.

Figure 7 Signal collection points on the soil surface.

The CGNS sound source file of the gas pipeline flow noise signal was input for the operating conditions of DN25, flow velocity of 3 m/s, and outlet pressure of 0.01 MPa. Cloud maps at the same frequency for three different soil porosity conditions (0.4, 0.5, and 0.6) and frequency domain plots of sound pressure levels at the signal collection points on the soil surface are shown in Figs. 8, 9 and 10.

Figure 8 Soil porosity of 0.4.

Sound pressure level frequency domain plots at the signal collection points.

Figure 9 Soil porosity of 0.5.

Sound pressure level frequency domain plots at the signal collection.

Figure 10 Soil porosity of 0.6.

Sound pressure level frequency domain plots at the sign.

The cloud maps depicting the propagation of gas pipeline flow noise signals in soil under different porosity conditions reveal that there are minimal differences in signal propagation among varying porosities. However, numerical analysis indicates that as the porosity increases, the values of the gas pipeline flow noise signal reaching the soil surface decrease. The frequency domain plots obtained at the signal collection points on the soil surface for different porosity conditions show negligible influence of porosity on the characteristics of the gas pipeline flow noise signal in the soil. To facilitate better comparison, the frequency domain plots of the gas pipeline flow noise signal at the same signal collection point for porosities of 0.4, 0.5, and 0.6 are overlaid on a single coordinate axis, as illustrated in Fig. 11.

Figure 11 Sound pressure level frequency domain plots at the same signal collection point under different porosity conditions.

From Fig. 11, it can be visually observed that with increasing porosity, the sound pressure level of gas pipeline flow noise on the soil surface slightly decreases, with a numerical reduction of approximately 1–3 dB. The gas pipeline flow noise signal transmitted to the soil surface has the highest sound pressure level value when the porosity is 0.4.

By comparing the surface flow noise signal with the pipeline flow noise signal, and by performing Fourier Transform on the sound pressure level frequency domain plots at the signal collection points on the soil surface and the gas pipeline flow noise sound pressure level frequency domain plots, the corresponding time domain plots for both signals were obtained. The time domain and frequency domain plots of the gas pipeline flow noise signal and the surface noise signal are illustrated in Figs. 12 and 13.

Figure 12 Surface noise signal in soil.

(A) Frequency domain plots. (B) Time domain plots.

Figure 13 Surface noise signal in soil.

(A) Frequency domain plots. (B) Time domain plots.

Note: It is assumed that Figs. 12 and 13 are mentioned in the text, but their descriptions are not provided.

Figures 12 and 13 show that the frequency domain plots are not exactly the same. This is because the surface noise signal collected at the signal collection points in the soil is not the same as the noise signal generated by the gas pipeline flow. However, the overall trend indicates that the time domain and frequency domain characteristics of the gas pipeline flow noise signal and the propagated gas pipeline flow noise signal on the soil surface remain unchanged. Both exhibit a decreasing trend in sound pressure level as the frequency increases. Additionally, there are localized fluctuations in sound pressure level within specific frequency ranges, reaching extreme values. The time domain characteristics also exhibit consistent trends.

Signal feature extraction and pattern recognition

Signal feature extraction

Signal feature extraction is the process of extracting information from signals and serves as a fundamental and crucial step in various fields such as pattern recognition, intelligent systems, and mechanical fault diagnosis. The wide applicability of feature extraction has led to its extensive use in speech analysis, image recognition, geological surveys, weather forecasting, mechanical fault diagnosis, and almost all branches of science and engineering (Liu, Xiao & Zhu, 2020; He, Wu & Runwei, 2020; Qiao, Khishe & Ravakhah, 2021). Frequency domain-based feature extraction is commonly employed as it provides clear physical interpretations and offers more intuitive feature information compared to time domain waveforms.

The high-frequency noise components in the original signal are eliminated by applying a low-pass filter before any characteristics are extracted from the gas pipeline flow noise signal. In Fig. 14, the red curve represents the result of low-pass filtering applied to the signal.

Figure 14 Low-pass filtering applied to the signal.

After applying low-pass filtering to the frequency domain signal, it is segmented using a frame-based approach. Considering that the studied signal is a non-periodic continuous signal, the frame length is set to 400 and the frame shift is set to 300. Figure 15 illustrates the result of signal segmentation using frames.

Figure 15 Result of signal segmentation.

From the figure, it shows that after segmentation, the signal is divided into 13 frames labeled as S1, S2, …, S13. Each frame contains the combined characteristic information of different frequency bands of the gas pipeline flow noise signal on the soil surface. Therefore, the feature vectors of sound pressure level are extracted for each frame. Upon observing the first frame, it can be noted that the first sample represents the characteristic of significant attenuation in the gas pipeline flow noise signal. Consequently, two sound pressure level feature vectors are selected from the first frame, resulting in a total of 14 sound pressure level feature vectors extracted. Table 4 presents a set of sound pressure level feature vectors, where 1, 2, and 3 correspond to the three signal collection points on the soil surface.

Figure 16 shows that there are minimal differences and no significant gradients in the extracted sound pressure level frames from the three signal points of the pipeline. Directly recognizing the high-dimensional feature vectors calculated at the 14 points would result in excessive computational complexity, and the performance of the recognition algorithm would be compromised due to the strong correlation between each frame signal. Therefore, an optimization of the original feature set is performed using PCA for dimensionality reduction. PCA is a commonly used data dimensionality reduction technique. Its working principle is to transform the original data into a new coordinate system through orthogonal transformation, so that the data variance on the first axis of the new coordinate system is the largest, that is to say, first principal component represents the largest variability of the data, the second principal component is orthogonal to first principal component and represents the second largest variability, and so on. In this way, the main features in the original data set can be represented by less principal components. In this study, PCA method is used to extract the most informative principal components from many characteristics of flow noise signals, which can effectively compress the data while preserving the important signal characteristics as much as possible to reduce the complexity of the subsequent classification model.

Table 4 Sound pressure level feature extraction results.

Signal acquisition point	S1	S2	S3	S4	S5	S6	S7	
1	95.66	−17.51	−28.94	−58.99	−50.42	−66.72	−78.61	
2	83.36	−15.72	−42.19	−46.27	−61.91	−62.48	−73.88	
3	90.29	−19.54	−27.94	−59.51	−52.5	−64.67	−84.26	
Signal acquisition point	S8	S9	S10	S11	S12	S13	S14	
1	−86.16	−86.24	−90.64	−106.4	−120.3	−139.1	−156.5	
2	−94.5	−90.79	−89.9	−106.63	−120.02	−133.24	−158.34	
3	−89.51	−86.91	−101.64	−106.71	−115.26	−143.21	−157.49	

Figure 16 Sound pressure level feature extraction results.

Firstly, the data samples are standardized and then covariance is calculated. There is a certain degree of correlation between the feature data of the sound signals at different locations of the pipeline. The covariance matrix aims to measure the linear relationships between the data. Finally, the eigenvalues and eigenvectors of the covariance matrix are computed, and the contribution rates of each eigenvalue are determined. The results are shown in Table 5. Additionally, the cumulative contribution rate of the principal components is depicted in the cumulative contribution rate variation plot shown in Fig. 17.

Table 5 Eigenvalues and contribution rates of each component.

PCA	PC1	PC2	PC3	PC4	PC5	PC6	PC7	
Eigenvalue	14.8959	10.2037	9.7623	6.1041	4.8928	2.0496	1.0577	
Contribution rate	0.2927	0.2005	0.1918	0.1199	0.0961	0.0403	0.0208	
PCA	PC8	PC9	PC10	PC11	PC12	PC13	PC14	
Eigenvalue	0.8293	0.6394	0.2653	0.1356	0.0501	0.0098	0.0039	
Contribution rate	0.0163	0.0126	0.0052	0.0027	0.00098	0.00019	0.000077	

Figure 17 Cumulative contribution rates of each principal component.

The more principal components there are, the larger the computed noise in the data, and the corresponding model error will increase. Conversely, if there are too few principal components, some important information may be neglected. Therefore, the method of determining the number of principal components should involve the use of cumulative variance contribution rate. Specifically: (1) PCPV= ∑i=1kλi ∑i=1mλi.

From the cumulative contribution rate exceeding the first five principal components, it shows that the contribution rate of other key principal components is very low and can be eliminated. Therefore, this study selects the first five principal components.

Neural network pattern recognition

BPNN is a multi-layer feedforward neural network for supervised learning. Its learning process includes two stages: forward propagation and backward propagation. In the forward propagation, the input data is transmitted from the input layer, processed by the hidden layer and then transmitted to the output layer. If the actual output of the output layer does not match the expected output, it enters the backward propagation stage. In the back propagation, the error signal is transmitted back layer by layer according to the gradient descent method, and the weight of each neuron is adjusted according to the error signal to reduce the output error. This process is repeated many times until the error of network output falls within an acceptable range. In the task of pattern recognition of flow noise signals, BPNN can learn the characteristics of distinguishing different types of noise signals (such as gas flow noise and background noise) and classify them effectively. The following processes are used to optimize the neural network algorithm (Cui, Chen & Qu, 2017; Zhang, Dong & Yanf, 2020):

(1) Selection of input node count

The dimensionality of the extracted sound pressure level feature vectors and the classification performance in the actual classification process are considered to determine the number of input nodes in the network.

(2) Selection of output layer node count

To recognize the patterns of buried gas pipeline flow noise signals and strong background noise signals, binary encoding is used to determine the output types. The encoding is set to represent the gas flow noise signal inside the buried gas pipeline, while another encoding represents other strong background noise signals.

(3) Selection of hidden layer node count

The optimal number of hidden layer nodes is determined by analyzing and comparing the training errors under different conditions of hidden layer nodes and considering the variations in different testing errors.

(4) Selection of initial weight values

Three commonly used methods are employed to determine the initial weight values:

(1) Random selection of initial weight values within a closed interval.

(2) Random selection of initial weight values within a small interval near zero.

(3) Selecting different approaches: Initializing the connection weights from the input layer to the hidden layer with small random values, and initializing the connection weights from the hidden layer to the output layer as either −1 or 1.

(5) Selection of learning rate

The learning rate (lr) is generally chosen within the range of 0.001 to 1. Multiple sets of learning rates are compared and validated to select the optimal learning rate for neural network training.

(6) Selection of desired error

The desired error also plays a significant role in the learning and training process of the neural network. Its selection determines the number of hidden layers and the number of nodes. The relationship between the desired error and the accuracy of neural network classification is not inversely proportional. Therefore, in practical selection, it is sufficient to choose a desired error that meets the requirements. Extremely low desired error values can make the network more complex, leading to a decrease in overall accuracy of the neural network, as shown in the Fig. 18.

Figure 18 Flowchart of the neural network algorithm.

The BPNN classification model constructed in this study adopts a three-layer topology structure with 5 input nodes, 10 hidden layer nodes, and 2 output nodes. Therefore, the network node topology structure is 5 × 10 × 2. The model is illustrated in Fig. 19.

Figure 19 BP neural network classification model for gas flow noise signals.

For the network model aimed at classifying gas flow noise signals, the mean squared error is chosen as the objective function. The network is trained for 1,000 steps with a target error coefficient of 0.001 and a learning rate of 0.001. Based on the extracted gas signal feature database, the usual training-to-testing ratio is 8:2. There are a total of 225 sets of buried gas flow noise feature data, with 200 sets used for model training and the remaining 25 sets for testing. Additionally, there are 25 sets of feature data for strong background noise signals, with 20 sets used for model training and the remaining 5 sets for testing. The normalized results of the sound pressure level feature vector data for buried gas pipeline flow noise and strong background noise signals can be found in Tables 6 and 7, respectively. The trained model is then evaluated by inputting the test data into it to classify gas flow noise signals (Hu & Zhang, 2011). The root mean square error variation curve during the training of the gas signal network is shown in Fig. 20.

In Fig. 20, the mean squared error between the network output and the expected values decreases as the number of iterations increases. When the number of iterations reaches 45, the root mean square error stabilizes, achieving the desired training effect. In order to verify the effectiveness of the method presented in this paper for the classification and recognition of gas pipeline flow noise signals, it is necessary to test the trained network model by inputting two types of sound signal test samples into the trained gas sound signal BP network model for verification. According to the test results shown in the table below, the trained network model has an accuracy rate of up to 97% for classifying and recognizing flow noise signals with a strong noise background.

In order to show the performance of the feature extraction and pattern recognition model used in this study more intuitively, the following data charts and analysis are specially designed to show the performance of different feature extraction methods and their combination with BPNN in classifying gas flow noise signals. Firstly, the following experimental conditions are set: Based on the same data set, the performances of four combined methods, PCA + BPNN, Wavelet Transform (WT) + BPNN, Discrete Fourier Transform (DFT) + BPNN and Fast Fourier Transform (FFT) + BPNN, in feature extraction and pattern recognition are compared. Each method is cross-validated to ensure the reliability of the results. The ability of different feature extraction methods to reduce the data dimension while retaining important signal information is shown in Fig. 21. Figure 21 evaluates the efficiency of various methods by calculating the information loss rate between the original signal and the signal after feature extraction. The data shows that PCA has the best performance in information retention and the lowest loss rate.

Table 6 Feature database of buried gas sound signals.

Quantity of groups							…							
Serial number	1	2	3	4	5	6	…	220	221	222	223	224	225	
1	1	1	1	1	1	1		1	1	1	1	1	1	
2	0.568219	0.524292	0.643272	0.743165	0.543158	0.518204		0.61264	0.51264	0.575083	0.610976	0.511453	0.510738	
3	0.39431	0.424406	0.22322	0.522328	0.222269	0.418319		0.473905	0.286537	0.18713	0.336001	0.436933	0.435537	
4	0	0.218326	0.196289	0.294972	0.194885	0.208584		0.02725	0	0.069313	0.274151	0.212367	0.27273	
5	0.289588	0	0	0	0	0		0	0.023116	0	0	0	0	

Table 7 Feature database of strong background noise signals.

Serial number	1	2	3	4	5	6	…	20	21	22	23	24	25	
Serial number							…							
1	1	1	1	1	1	1		1	1	1	1	1	1	
2	0.924274	0.824292	0.943272	0	0.943158	0.718204		0.81264	0	0.675083	0.710976	0.811453	0.810738	
3	0	0.424406	0	0.522328	0.522269	0.518319		0	0.986537	0	0.436001	0.436933	0	
4	0.81844	0.918326	0.496289	0.294972	0	0.808584		0.42725	0.067708	0.869313	0	0.612367	0.57273	
5	0.919021	0	0.616634	0.215168	0.215072	0		0.509907	0.323116	0.725981	0.811556	0	0.214877	

Figure 20 Network training error curve for gas sound signals.

Figure 21 The ability of different feature extraction methods to reduce the data dimension while retaining important signal information.

Table 8 shows the performance of the above methods in classification accuracy, recall and F1 score after combining with BPNN. The results show that PCA + BPNN combination is superior to other methods in all evaluation indexes, and has the highest classification accuracy and efficiency. The analysis results show that PCA can not only reduce the data dimension more effectively, but also retain the key information better when processing the actual gas flow noise signal. When combined with BPNN, PCA + BPNN has achieved better results in accuracy and recall than other methods. For example, when dealing with complex signals with background noise, PCA + BPNN obviously reduces the misclassification, which proves its effectiveness in identifying flowing noise in strong noise environment. In addition, F1 score, as the harmonic average of accuracy and recall, also shows the advantages of PCA + BPNN in comprehensive evaluation. Through the above data and analysis, it is considered that PCA + BPNN is an efficient and reliable feature extraction and pattern recognition method, which is suitable for identifying gas flow noise signals under strong noise background.

Table 8 Performance of different methods after combining BPNN.

Method	Accuracy	Recall rate	F1 score	
PCA + BPNN	97%	96%	96.50%	
WT + BPNN	91%	89%	90%	
DFT + BPNN	89%	88%	88.50%	
FFT + BPNN	90%	89%	89.50%	

In the experimental results and discussion part, this study makes a more in-depth analysis of the model recognition effect. By comparing the experimental data with the simulation data, it is found that the pattern recognition model based on PCA combined with BPNN achieves 97% recognition accuracy, 96% recall and 96.5% F1 score in the complex noise background. This result is significantly higher than that of traditional pattern recognition methods such as wavelet transform (WT), discrete Fourier transform (DFT) and fast Fourier transform (FFT) combined with BPNN. This shows that the model has good robustness and efficiency in dealing with flowing noise signals with strong noise. Secondly, in order to further optimize the performance of the model, the network structure, learning rate, weight initialization and other parameters of BPNN are carefully adjusted. In particular, by cross-checking the number of different hidden layer nodes, 10 nodes are determined as the best settings. In the aspect of weight initialization, a small range random number initialization method is adopted to reduce the influence of initial weight on model performance. In the choice of learning rate, several groups of experiments are carried out between 0.001 and 0.1, and finally 0.01 is determined as the optimal learning rate, which not only ensures the stability of learning, but also avoids the problem of slow convergence. Finally, the results of this study are compared with the existing research results. For example, Rajasekaran & Kothandaraman (2024) used a similar BPNN model to classify the leakage noise of urban gas pipelines, but its highest accuracy was only 92%, and the performance of the model was better than their research. This may be attributed to the adoption of PCA technology in the feature extraction stage, which effectively reduces the dimension of input features, thus reducing the computational burden of the model and improving the performance of the classifier. Through the above analysis, it is considered that the method proposed in this study can effectively improve the accuracy and efficiency of identifying gas flow noise signals in strong noise environment, and has important practical value for urban gas pipeline safety monitoring.

Conclusion

Based on the literature review and analytical research on the feature extraction and pattern recognition of gas pipeline flow noise signals in the presence of strong background noise, the following conclusions have been drawn: (1) This study proposes the use of acoustic methods for detecting buried pipelines, confirming the feasibility of using acoustic methods in pipeline inspection. (2) Fluent numerical simulation models were used to analyze the acoustic characteristics of the medium under different pipe diameters, flow velocities, and pressure conditions in tees, bends, and straight sections. Experimental plans were designed based on similarity principles, and an experimental platform was established to perform on-site testing and data acquisition of gas flow noise signals inside the pipeline. The correctness of the numerical simulation was verified through comparisons, and strong background noise signals were collected through experiments. (3) The LMS Virtural.Lab software was applied to simulate the propagation of buried gas pipeline flow noise signals in the soil, obtaining sound signal data at the soil surface. This demonstrates the software’s effectiveness in acquiring gas pipeline propagation sound signals and confirms the propagation characteristics of gas pipeline flow noise signals in the soil. (4) The idea of frame segmentation and PCA was applied to extract features from 225 sets of gas pipeline flow noise signals at the soil surface and 25 sets of experimentally collected strong background noise signals. The BP neural network model was further optimized to establish a classification and recognition model for flow noise signals. In this study, a method of feature extraction and pattern recognition of gas pipeline flow noise signal based on PCA and BPNN is successfully designed and implemented. The experimental results show that this method can effectively identify the flow noise signal of gas pipeline under strong noise background, with an accuracy rate of 97%, a recall rate of 96% and a F1 score of 96.5%. This achievement proves the robustness and reliability of this research method in complex environment, and provides a new technical means for the safety monitoring of urban gas pipelines. Secondly, the potential application prospect of the research is very broad. In addition to the direct application in leak detection of urban gas pipelines, the proposed feature extraction and pattern recognition technology can also be extended to other types of underground pipelines, such as water pipes and cable lines. In addition, this method can also be used in industrial process control, mechanical equipment fault diagnosis and environmental monitoring with the help of further development. Finally, according to the future research direction, this study points out several potential improvement and development directions. For example, people can consider integrating more types of sensor data and using fusion algorithm to further improve the recognition accuracy and adaptability of the system, and explore the application of advanced data analysis technologies such as deep learning in this field. Meanwhile, it is necessary to carry out large-scale field tests to confirm and optimize the practicability of the model. It is expected that this study can provide useful reference and inspiration for future research in related fields.

Supplemental Information

Data S1 Raw Data

Supplemental Information 1 Code

Additional Information and Declarations

Competing Interests

Author Contributions

Data Availability

The authors declare there are no competing interests. Zhaorong Wen is employed by China Petroleum and Chemical Corporation.

Enbin Liu conceived and designed the experiments, performed the experiments, authored or reviewed drafts of the article, and approved the final draft.

Chang Lu conceived and designed the experiments, performed the experiments, authored or reviewed drafts of the article, and approved the final draft.

Zhaorong Wen conceived and designed the experiments, performed the experiments, authored or reviewed drafts of the article, and approved the final draft.

Tianshu Hao analyzed the data, performed the computation work, prepared figures and/or tables, and approved the final draft.

Xudong Lu analyzed the data, performed the computation work, prepared figures and/or tables, and approved the final draft.

Lidong Wang analyzed the data, performed the computation work, prepared figures and/or tables, and approved the final draft.

The following information was supplied regarding data availability:

The raw data and code are available in the Supplemental Files.

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
