# Peer review of "Feature extraction and pattern recognition of gas pipeline flow noise signals in a strong noisy background"

_PeerJ Computer Science, doi:10.7717/peerj-cs.2087_

## Round 0.1 · original submission · Major Revisions

Please improve your manuscript according to the comments of the reviewers.

**Language Note:** The review process has identified that the English language must be improved. PeerJ can provide language editing services - please contact us at [email protected] for pricing (be sure to provide your manuscript number and title). Alternatively, you should make your own arrangements to improve the language quality and provide details in your response letter. – PeerJ Staff

Reviewer 1 ·

Basic reporting

1. In the abstract section of the article, the discussion on the research background is too long to display the research purpose and motivation. Please briefly state the research purpose at the beginning of the abstract. Additionally, the abstract lacks research conclusions, so the author is requested to supplement.
2. The introduction to the application fields and software tools of acoustics mentioned in the introduction section is very helpful, but more detailed information is needed on how to choose specific tools, including why to choose LMS Virtual Lab software and its advantages compared to other software.
3. There is a lack of literature review section. It is necessary to elaborate on the existing gas pipeline flow noise signal feature extraction methods and explain the problems to be solved in this field.
4. When introducing models and algorithms, it is necessary to provide a deeper explanation of the working principles of PCA and BP neural networks to help readers understand how these methods are applied to feature extraction and pattern recognition of flow noise signals.

Experimental design

1. In the data and experimental methods mentioned in the article, more technical details should be provided, such as the specific construction of the experimental platform and the characteristics of the sensors used, so that readers can understand the repeatability and reliability of the experiment.
2. The test results of model performance should be displayed with relevant data charts, and need to be compared horizontally with classic feature extraction methods.

Validity of the findings

1. In the results and discussion section, the authors can conduct a more in-depth analysis of the experimental results and the performance of the model, and compare them with existing research results. More detailed information about the model optimization process and algorithm adjustments can also be provided.
2. The conclusions proposed in the conclusion section need to be more specific, which can further clarify the significance and potential application prospects of the study, as well as provide guidance for future related research.

·

Basic reporting

The paper at this level of elaboration has satisfied all points inserted in the guidance page as well. I agree with all of them

Experimental design

also in thi step I am fully satisfied. I consider the paper in this formulation entirely falls into the aim and the scope of the Journal. No comment for the others.

Validity of the findings

No comments. The impact and the novelty are clear and well defined. The scientific sound is high

Additional comments

None

Reviewer 3 ·

Basic reporting

The work could be interesting. More research background and critical discussion are expected. Significant improvement is required.
1.More research background of different feature extraction and classification are required e.g.
Ali Imam Sunny, etc, Low frequency (LF) RFID sensors and selective transient feature extraction for corrosion characterisation, Sensors and Actuators A: Physical, Volume 241, 2016, Pages 34-43, ISSN 0924-4247, https://doi.org/10.1016/j.sna.2016.02.010.
A. A. Khlybov, A. L. Uglov, T. A. Bakiev & D. A. Ryabov (2023) Assessment of the degree of damage in structural materials using the parameters of structural acoustic noise, Nondestructive Testing and Evaluation, 38:2, 331-350, DOI: 10.1080/10589759.2022.2126470

Experimental design

More evaluation and validation are required.

Validity of the findings

More cpmparison are expected.

Reviewer 4 ·

Basic reporting

The writing language and the resolution of figures can be improved.

Experimental design

No comment.

Validity of the findings

The contribution of the presented work is limited.

Additional comments

This manuscript presents a pipeline flow noise signal detection method through feature extraction and neural network. Overall, the novelty and contribution of the presented work is limited. The manuscript reads like a technical report rather than a scientific article. Significant revisions are needed before the publication.
(1)What is the main challenge in the field of gas pipeline flow noise signal detection? The contribution of the presented work compared to existing works should be highlighted.
(2)BPNN is a very common machine learning tool for the pattern recognition problem. The novelty of the presented work is limited unless it develops state-of-the-art algorithms or major improvements to existing methods.
(3)PCA is a common feature extraction method. Is the PCA the optimal method for extracting features from the flow noise signals?
(4) The figures in the manuscript suffer from poor resolution, which affects their clarity and readability.
(5) Numerous typos are present throughout the manuscript, which detracts from the overall professionalism and clarity of the writing.
(6) The writing language can be improved for better readability.

---

## Round 0.2 · accepted · Accept

Although Reviewer 3 gave a suggestion the comments were very general. So I don't think the comments from Reviewer 3 have much value.

·

Basic reporting

I confirm my decision on

Experimental design

I confirm my decision on

Validity of the findings

I confirm my decision on

Reviewer 3 ·

Basic reporting

It is one timely paper.

Experimental design

.

Validity of the findings

.

Additional comments

More comparison and critical discussion are required.